# Structural Insights into the Regulatory Mechanisms of the Toxic Activity of Sofic in Anti-Phage Defense Systems

**DOI:** 10.3390/ijms26136074

**Published:** 2025-06-24

**Authors:** Zhuoxi Wu, Guodong Chen, Libang He, Hao Guo, Ruifang Yuan, Huiling Su, Zhenyang Xie, Faxiang Li

**Affiliations:** MOE Key Laboratory of Rare Pediatric Diseases, Center for Medical Genetics, School of Life Sciences, Central South University, Changsha 410013, China; wzx0326@csu.edu.cn (Z.W.); guodong.chen@csu.edu.cn (G.C.); heboercsu@csu.edu.cn (L.H.); 232511008@csu.edu.cn (H.G.); 242511041@csu.edu.cn (R.Y.); suhuiling@csu.edu.cn (H.S.); 2804230121@csu.edu.cn (Z.X.)

**Keywords:** AMPylation, anti-phage defense, toxin–antitoxin, abortive infection

## Abstract

The FIC domain-containing protein Sofic has recently been shown to provide robust protection to bacteria against phage infection. Sofic acts as a toxic protein, inducing abortive infection through the AMPylation of target proteins during phage invasion. However, the molecular mechanisms regulating Sofic’s toxic activity remain elusive. In this study, we identified a small gene encoding a short protein located downstream of *Sofic* in the genome, named AS1 (anti-Sofic1), which functions as an antitoxic protein to counteract Sofic’s toxicity. The crystal structure of Sofic revealed that the protein functions as a dimer in solution, with dimerization being indispensable for its toxic activity. Importantly, structural analysis indicated that ATP binding induces a conformational change in the C-terminal domain (CTD) of Sofic, underscoring the critical role of the CTD in mediating its toxic effects. In vitro colony-forming assays confirmed that the interaction between the CTD and the Amylase domain is crucial for Sofic’s toxic activity. Overall, our results provide molecular insights into the regulatory mechanisms of Sofic in antiviral immunity.

## 1. Introduction

To combat phage infection, bacteria have evolved hundreds of sophisticated anti-phage defense systems to prohibit phage propagation [1,2,3]. Most of these systems, such as the restriction–modification (RM), CRISPR-Cas, Shedu, and tail assembly blocker (Tab) systems, inhibit phage infection by directly targeting phage components, degrading phage DNA, or blocking tail tube assembly [4,5,6]. However, a growing number of recently identified systems mediate phage immunity using the abortive infection method by inducing cell death in infected bacterial cells to inhibit phage reproduction [7]. Examples of these systems include the toxin–antitoxin, CBASS, Sir2-HerA, Thoeris, and defense-associated sirtuin (DSR) systems [8,9,10,11,12].

Toxin–antitoxin (TA) systems are prevalent genetic modules in bacteria, integral to their survival and stress response mechanisms [13,14]. These systems comprise a toxic protein that impairs vital cellular functions and an antitoxic protein that neutralizes the toxin’s effects under normal conditions. Upon phage infection, the antitoxin’s inhibitory action is disrupted, releasing the toxic protein and often inducing cell death or dormancy in the infected cell, thereby curbing phage dissemination [15]. TA systems are categorized into eight distinct types based on the molecular mechanisms by which the antitoxin counteracts the toxin, ranging from direct protein sequestration to RNA-mediated regulation. Among these, proteins containing FIC (filamentation induced by cyclic adenosine monophosphate) domains function as part of TA-like modules, playing a pivotal role in bacterial immunity [16,17]. FIC proteins are a family of enzymes known for catalyzing adenylylation (also called AMPylation)—the transfer of AMP onto a target protein—and are found in both prokaryotic and eukaryotic organisms [18,19,20]. The injection of FIC proteins from pathogenic bacteria into mammalian cells leads to the adenylylation of small GTPases, which disrupts downstream signaling and ultimately results in cell death [21]. For example, *Vibrio parahaemolyticus* VopS10 and *Histophilus somni* IbpA11 trigger actin cytoskeleton collapse and cell death by mediating the adenylylation of Rho family GTPases.

Structural studies indicate that the toxic effects of FIC proteins are suppressed by the inhibitory α-helix (α_inh_). These proteins can be categorized into three classes based on whether the α_inh_ originates from the antitoxin proteins (class 1), or the N-terminal (class 2) or the C-terminal (class 3) helix of the FIC domains [16,22,23]. Recent bioinformatics analyses and plaque assays indicate that the class 2 family of FIC proteins, known as Sofic, functions as a single-protein anti-phage defense system, protecting bacteria from phage infection through unknown mechanisms [24]. Although Sofic is expected to be in an auto-inhibited state due to the N-terminal α_inh_, the failure to make the wide-type construct for recombinant protein purification suggests that its toxic activity is active to some extent [23]. The molecular mechanisms regulating Sofic’s toxic activity still require further investigation.

Here, we first identified a 47-residue protein named AS1, located downstream of *Sofic* in *Escherichia coli* H397, which functions as the antitoxin that neutralizes the toxic effects of Sofic. Subsequent structural analyses and in vitro colony-forming assays reveal that dimerization and the interaction between the CTD and the AMPylase domain are critical for maintaining the toxic activity of Sofic. Our study not only uncovers the regulatory mechanisms of Sofic’s toxic activity but also provides a template for understanding the roles of FIC proteins in anti-phage defense processes.

## 2. Results

### 2.1. The Small Protein AS1 Counteracts the Toxic Effects of Sofic in E. coli

The previous study demonstrated that wild-type Sofic exists in an auto-inhibited state due to the presence of the inhibitory α-helix, α_inh_, within the catalytic pocket. However, the structure of Sofic from the bacterium *Shewanella oneidensis* MR-1(*So*Sofic) is that of the G109C mutant, not the wild-type protein [23]. In fact, we were also unable to clone the wild-type *Sofic* from *Escherichia coli* H397 (*EcSofic*) into the prokaryotic expression vector containing only one lac operator, which suggests the toxic effects of Sofic. To analyze the toxic activity of wild-type *Sofic*, we successfully constructed a prokaryotic expression plasmid for *EcSofic* under three contiguous lac operators to avoid leaky expression. This phenomenon suggests that the wild-type Sofic is an active toxic protein. To investigate the inhibitory mechanism of *Ec*Sofic, we analyzed the genome sequence and identified a small gene encoding 47 amino acids located downstream of *EcSofic*. This protein may function as an antitoxic agent against *Ec*Sofic, and we have named it *Ec*AS1 (anti-Sofic1) (Figure 1A). Subsequent in vitro colony-forming assays were employed to evaluate the toxic activity of *Ec*Sofic by transforming the prokaryotic expression constructs into the *E.coli* BL21-AI, which naturally lacks the *Sofic* gene. Compared to the transformation of the empty vector or the plasmids harboring the catalytically dead mutant H194A, the number of colony units formed by bacteria expressing the wild-type *Ec*Sofic is dramatically decreased (Figure 1B). In contrast, the colony units of the bacteria expressing the *Ec*Sofic G106C mutant (corresponding to *So*Sofic G109C) fall in between (Figure 1B). These results suggest that wild-type Sofic is an active toxic protein, while the G106C mutant exhibits reduced activity.

To test whether *Ec*AS1 could inhibit the toxic activity of *Ec*Sofic, we co-expressed *Ec*Sofic and *Ec*AS1 in *E. coli* BL21-AI using the pRSF-duet vector, which contains two multiple cloning sites (MCSs) for the co-expression of two proteins within a single vector, and conducted colony-forming assays. The results show that, similar to the expression of the *Ec*Sofic H194A mutant alone, the colony-forming units of bacteria co-expressing *Ec*AS1 and *Ec*Sofic exhibit a remarkable increase compared to those expressing *Ec*Sofic G106C alone, regardless of whether *EcSofic* is cloned into the first MCS with *EcAS1* in the second MCS (Sofic + AS1) or vice versa (AS1 + Sofic) (Figure 1C). To explore the inhibitory mechanism of AS1, we attempted to purify the *Ec*AS1 and *Ec*Sofic-*Ec*AS1 complex protein. However, we were unable to obtain soluble *Ec*AS1 or the *Ec*Sofic-*Ec*AS1 complex, and we also failed to purify the *Ec*Sofic protein alone from the co-expression system. It is possible that *Ec*AS1 represses the toxic effects of *Ec*Sofic by regulating its stability.

### 2.2. The Crystal Structure of Apo Sofic

To characterize the mechanism of *Ec*Sofic for AMPylation, we determined the crystal structure of the *Ec*Sofic H194A mutant at a resolution of 2.3 Å (Table 1). There are two *Ec*Sofic molecules in the asymmetric unit, forming a homodimer. The structure shows that *Ec*Sofic consists of 15 α-helices and 3 β-strands, folding into two domains: the N-terminal AMPylase domain (AD) and the C-terminal CTD (Figure 2A). The AD domain of *Ec*Sofic is composed of α1 to α11, while the CTD consists of the remaining three α-helices and three β-strands (Figure 2A). We also co-crystallized the *Ec*Sofic H194A mutant protein with the substrate AMPPNP (an ATP analogue), but no densities corresponding to AMPPNP were observed in the final structure. The H194A mutation may disrupt Sofic’s binding to the substrate. Alternatively, we superimposed our apo *Ec*Sofic structure with the ATP-bound form of *So*Sofic from *Shewanella oneidensis* to elucidate the mechanism of substrate coordination. The overlaid structure shows that the AMPylase domains of the apo *Ec*Sofic and ATP-bound forms are identical, while the CTD in the apo *Ec*Sofic dimer exhibits an outward twist (Figure 2B). This indicates the critical role of the CTD in regulating Sofic’s toxic activity. The key residues (E73, R206, H198, and Y241) of *So*Sofic that directly participate in ATP coordination in the catalytic pocket are highly conserved in *Ec*Sofic (E70, R202, H194, and Y237) (Figure 2C). The subsequent colony-forming assay unveiled that mutation of any of the key residues (R202A or Y237A) within the catalytic pockets dramatically decreased the toxic activity of *Ec*Sofic (Figure 2D).

### 2.3. Dimerization of Sofic Is Required for Its Toxic Activity

The crystal structure of apo *Ec*Sofic indicates that Sofic may function as a homodimer in solution. The structure shows that dimerization of *Ec*Sofic is mediated by the α1 helices of the two molecules in the *Ec*Sofic dimer, which form a coiled-coil structure. Two major interfaces were observed in the *Ec*Sofic dimer, with interactions mediated by both hydrophilic and hydrophobic contacts (Figure 3A). At interface 1, the positively charged side chains of R25 and K29 form extensive salt bridges and hydrogen bonds with the negatively charged side chains of E23’ and D176’ from the neighboring protomer. The hydrophobic side chains of I32 from both *Ec*Sofic molecules are in close proximity, creating hydrophobic interactions that strengthen the dimer interface (Figure 3B). At interface 2, the hydrophilic side chains of D40 and T43’ form hydrogen bonds, while the hydrophobic residues L47 from both sides establish a hydrophobic interaction to further stabilize the dimer complex (Figure 3C). The oligomerization state of *Ec*Sofic in solution was analyzed using analytical ultracentrifugation (AUC). The measured molecular weight of *Ec*Sofic is 81.3 kDa, which is very close to the theoretical dimer size of 82 kDa, further confirming the dimerization of *Ec*Sofic in solution (Figure 3D).

To investigate the impact of *Ec*Sofic dimerization on its toxic activity, we first mutated key residues in the dimer interface to generate monomeric *Ec*Sofic. The analytical ultracentrifugation (AUC) results indicate that the R25E and I32E single mutants of *Ec*Sofic disrupt the dimer, which have molecular weights of 41.3 kDa and 41.8 kDa, respectively (Figure 3D). Subsequently, colony-forming assays were performed to analyze whether dimerization is critical for maintaining the toxic activity of *Ec*Sofic. The results reveal a robust increase in colony-forming units for bacteria transformed with the R25E and I32E mutants of *Ec*Sofic compared to the wild-type *Ec*Sofic, demonstrating that dimerization is indispensable for *Ec*Sofic’s toxic effect (Figure 3E).

### 2.4. The C-Terminal Domain of EcSofic Regulates Its Toxic Activity

The outward twist of the CTD in the apo *Ec*Sofic, compared with the ATP-bound *So*Sofic, suggests a critical role for the CTD in regulating the AMPylase activity of Sofic (Figure 2B). Structural analysis reveals that the CTD interacts with the AMPylase domain (AD) through two major interfaces. The first interface is formed by α15 of the CTD and α11 of the AMPylase domain. The hydrophobic side chain of L352 from the CTD inserts itself into a hydrophobic pocket formed by I273, V274, L277, and I57 of the AMPylase domain. In addition, a salt bridge between R351 of the CTD and E280 from the AMPylase domain was also observed, further strengthening the interaction (Figure 4A). The second interface is formed by β1 of the CTD and α3 of the AMPylase domain. In this interface, the phenolic hydroxyl group of Y306 on β1 establishes a hydrogen bond with the hydrophilic side chain of D77 on α3. Moreover, R308 of the CTD constitutes two hydrogen bonds with the side chain and main chain of Q81 from the AMPylase domain (Figure 4B). The subsequent colony-forming assays demonstrate that deletion of the CTD (P290X) or α15 (T350X), or the L352E mutation, abolished the toxic effect of *Ec*Sofic, exhibiting a remarkably increased number of colony-forming units (Figure 4C). The single mutations Y306A and R308A on the second interface largely alleviate the toxic effect of *Ec*Sofic, showing a moderate number of colony-forming units compared with expressing wild-type *Ec*Sofic (Figure 4D). The above results demonstrate that the CTD plays a pivotal role in regulating the toxic activity of *Ec*Sofic.

## 3. Discussion

Previously, the FIC domain-containing protein Sofic was discovered to be widely distributed in defense islands, functioning as a single-protein anti-phage defense system [25,26,27]. In bacteria, Sofic is thought to trigger abortive infection by AMPylating target proteins, leading to cell death and thereby preventing phage propagation. This mechanism is consistent with the broader role of FIC proteins in bacterial physiology and immunity, where they often function as toxins in toxin–antitoxin (TA) systems, contributing to stress response, persistence, and virulence [21,28,29]. Therefore, the toxic activity of Sofic must be precisely regulated—suppressed in uninfected cells and activated during phage infection.

The previous study revealed the auto-inhibition mechanism of Sofic in uninfected cells to minimize its toxic effects, while the activation mechanism of Sofic upon phage infection remains elusive [16]. However, our study, along with previous research, indicates that cloning the wild-type *Sofic* into a prokaryotic expression vector controlled by a single lac operator is challenging [23]. This suggests that the toxic effects of Sofic may be due to leaky expression and that the auto-inhibition mechanism does not fully inhibit Sofic’s activity.

In this study, we identified a small AS1, located on the defense island alongside *Sofic* in *Escherichia coli* H397, which functions as the antitoxin protein to neutralize the toxic effects of Sofic (Figure 1C). However, the precise mechanism by which AS1 interacts with Sofic remains unclear due to challenges in purifying the AS1-Sofic complex. This limitation underscores the need for future research to employ alternative purification strategies or structural biology techniques to elucidate this interaction. Understanding the molecular basis of AS1’s role could provide insights into how bacterial cells regulate Sofic’s activity to ensure it is suppressed in uninfected cells but activated during phage infection.

FIC proteins can be categorized into three groups based on the source of the inhibitory α-helix. Class 1 systems consist of two proteins that form toxin–antitoxin systems, with the inhibitory α-helix derived from the antitoxin proteins, such as *vbhT-vbhA*. In contrast, class 2 and class 3 systems feature single FIC domain-containing proteins, where the inhibitory α-helix originates from the N-terminal (class 2) or the C-terminal (class 3) helix of the domains, as seen in the *Sofic* and *NmFic* systems [16]. Our crystal structure reveals that *Ec*Sofic adopts a two-domain architecture consisting of an N-terminal AMPylase domain and a C-terminal CTD. The AMPylase domain folded into a class 2 FIC domain, featuring an inhibitory α-helix that originates from the N-terminal, despite the presence of the canonical catalytic core (Figure 2A). The mutation of key residues responsible for ATP coordination in the catalytic core results in the loss of toxic activity in *Ec*Sofic, further validating its AMPylation function (Figure 2D). In addition, the *Ec*Sofic functions as a homodimer in solution, and the dimerization was proven to be essential for its toxic activity (Figure 3). Despite the auto-inhibition and dimerization, the regulatory mechanism of *Ec*Sofic’s toxic activity appears to be more complicated due to the presence of the CTD. Disrupting the interaction between the CTD and the AMPylase domain by mutating key residues at the interface will inhibit *Ec*Sofic’s toxic activity (Figure 4). This precise control over toxic activity may make Sofic suitable for participation in anti-phage defense mechanisms.

Overall, this paper identifies Sofic-AS1 functions as a toxin–antitoxin system, highlighting that the interactions between the two domains and the two protomers in the *Ec*Sofic dimer are critical for its toxic activity. However, several key questions remain to be addressed. First, it is essential to identify the target protein of *Ec*Sofic in bacteria, as understanding this target will help uncover the mechanism of cell death induced by Sofic. Second, further investigation is needed into how AS1 neutralizes the toxic effects of Sofic. Given that toxin–antitoxin systems can be categorized into eight major types based on the nature of the antitoxin proteins [14], deciphering the inhibition mechanism is likely to be more challenging. Third, since the *Sofic* system has shown robust protection against T5 phage infection [24], additional phage-based screening assays should be conducted to explore the activation mechanism of Sofic during T5 phage infection. These efforts will not only deepen our understanding of bacterial defense strategies but also open new avenues for therapeutic interventions.

## 4. Materials and Methods


**Protein expression and purification**


The full-length genes of *Sofic* and *AS1* from *Escherichia coli* H397 (*EcSofic* and *EcAS1*) were synthesized at the company Tsingke. The complementary DNA fragment encoding the full-length *Ec*Sofic was PCR-amplified and cloned into the modified pETM.3C vector with three contiguous lac operators after the T7 promoter to avoid leaky expression. The H194A and G106C mutants of *Ec*Sofic were cloned into the pETM.3C vector (a modified version of the pET-32a vector that introduces an N-terminal 6xHis tag and a 3C protease cutting site before the multiple cloning site) for recombinant protein expression. The wide-type DNA fragments of *Ec*Sofic and *Ec*AS1 were cloned into the two cloning sites of the pRSF-duet vector for co-expression. All the plasmids used in this study were constructed using a Seamless Assembly Kit and performed according to the manufacturer’s protocol (ABclonal, Woburn, MA, USA, RK21020). All the point mutations of *EcSofic* used in this study were created using the standard PCR-based mutagenesis method, further checked by DNA sequencing.

The *Ec*Sofic H194A and G106C proteins were expressed in *Escherichia coli* BL21 (DE3) and BL21-AI bacterial cells, respectively. The bacteria were grown in LB medium with the appropriate antibiotic at 37 °C with shaking at 230 RPM. The cultivation temperature was lowered to 16 °C when the OD600 of the culture reached 0.8. Then, 200 µM IPTG (and 2 g/L arabinose for BL21-AI) was added into the bacterial cultures to induce protein expression, followed by overnight incubation at 16 °C.

The bacterial cell pellets were collected by centrifugation and then resuspended in five volumes of the binding buffer (50 mM Tris-HCl, pH 7.9, 500 mM NaCl, 8 mM imidazole). Subsequently, the cells were lysed by the ultrahigh-pressure homogenizer machine ATS-1500 (ATS engineering limited, Petah Tikva, Israel). The resulting cell lysate was spun down using a centrifuge at 35,000× *g* and 4 °C for 30 min to remove the pellets. The supernatant was carefully transferred into a new 50 mL centrifuge tube and mixed with 5 mL Ni^2+^-NTA agarose resin (Qiagen, Hilden, Germany, 30,230) that had been pre-equilibrated by the binding buffer. The mixture of supernatant and agarose resin was incubated at 4 °C for 1 h with rotation. After extensive washing with wash buffer (50 mM Tris-HCl, pH 7.9, 500 mM NaCl, 30 mM imidazole), the His_6_-tagged proteins were eluted from the Ni^2+^-NTA resin using a elution buffer (50 mM Tris-HCl, pH 7.9, 500 mM NaCl, 400 mM imidazole). The target proteins were further purified by size exclusion chromatography with the UNIONDEX 200PG column (Union Biotech, Shanghai, China). The pooled peak fractions were combined and concentrated and then stored at −80 °C for future use.


**Colony-forming assay**


The prokaryotic expression constructs containing the wild-type *EcSofic* or its various mutants were transformed into *E. coli* BL21-AI competent cells. The transformed cells were then plated and cultured in a 37 °C incubator for 14 h. Bacteria from a single colony were cultivated in 10 mL of LB medium at 37 °C with shaking for 6 h. The bacterial cultures were serially diluted 10-fold (from 10^0^ to 10^−7^), and 2 µL of each diluted bacterial culture was spotted onto an LB agar plate containing the appropriate antibiotic. After 14 h of culture in a 37 °C incubator, the bacterial colonies were recorded with a camera.


**Crystallography**


The *Ec*Sofic H194A protein was concentrated to 20 mg mL^−1^ in the buffer containing 20 mM Tris-HCl, pH 7.5, 100 mM NaCl, and 1 mM DTT and used for crystallization. Crystals of *Ec*Sofic were obtained through the sitting-drop vapor-diffusion method at 18 °C after 1 week by mixing 1 µL protein with an equal volume of a reservoir solution containing 0.1 M HEPES, pH7.5, 20% PEG1500, and 0.2 M L-Proline. Before the diffraction experiments were performed, glycerol (20%) was added as the cryo-protectant. High-resolution X-ray diffraction datasets were collected at 100 K at the beamline BL19U1 of the Shanghai Synchrotron Radiation Facility. The diffraction data were processed and scaled using HKL3000 [30].

The phase of the *Ec*Sofic H194A dataset was determined using the molecular replacement method with Phaser in Phenix [31,32], using the *Ec*Sofic structure model predicted by the alpha-fold3 online server as the search model. The structure models were iteratively manually adjusted to fit the electron density in COOT [33,34] and then refined using Phenix.refine. The qualities of the final model were verified by MolProbity [35]. The data collection and refinement statistics are summarized in Table 1.


**Analytical ultracentrifugation (AUC) assay**


We performed sedimentation velocity experiments using an Optima XL-1 analytical ultracentrifuge (Beckman Coulter, Brea, CA, USA) equipped with a four-cell rotor under 50,000 rpm at 12 °C. Samples were diluted to 0.8 mg/mL in the gel filtration buffer containing 20 mM Tris-HCl, pH 7.5, 100 mM NaCl, and 1 mM DTT. The buffer density was calculated using the program SEDNTERP (available at www.rasmb.org/) (accessed on 3 March 2015), and the final sedimentation velocity data were calculated using the software SEDFIT 16-1c.

## 5. Conclusions

In this work, we have identified AS1, a small protein that acts as an antitoxin, effectively counteracting the toxic effects of Sofic. Our crystal structure and colony-forming assays demonstrate that dimerization and the interactions between the C-terminal domain (CTD) and the AMPylase domain are both indispensable for the toxic effects of Sofic. These insights not only enhance our understanding of bacterial immunity but also pave the way for potential therapeutic applications targeting phage-resistant pathogens.

## Figures and Tables

**Figure 1 ijms-26-06074-f001:**
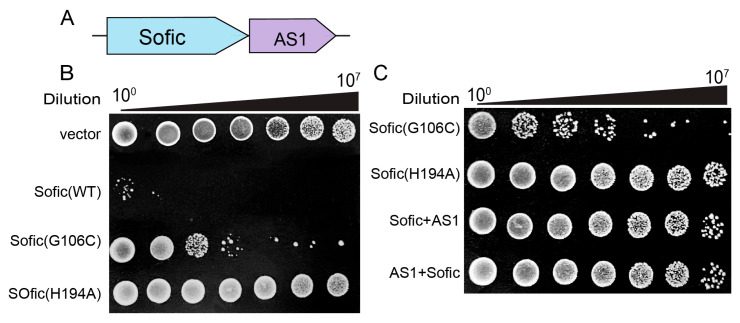
The toxic activity of *Ec*Sofic is repressed by *Ec*AS1. (**A**) Schematic diagram of *Sofic* anti-phage defense system. (**B**) Colony-forming assays showing the toxic activity of wild-type *Ec*Sofic and various mutants. The DNA fragments of *EcSofic* WT, G106C, or H194A mutants were cloned into the modified prokaryotic expression vector pETM.3C with three contiguous lac operators after the T7 promoter. The constructs were then transformed into *E. coli* BL21-AI to perform the colony-forming assays. The bacterial cultures from a single colony were serially diluted 10-fold after 6 h of cultivation. (**C**) Colony-forming assays showing that the toxic effect of *Ec*Sofic is repressed by *Ec*AS1. In this assay, the DNA fragments encoding G106C or H194A of *Ec*Sofic were cloned into the first multiple cloning site (MCS) of the prokaryotic expression vector pRSF-duet, which contains two MCSs for the co-expression of two proteins within a single vector. For the co-expression experiments, *EcSofic* and *EcAS1* were cloned into the two MCSs of the pRSF-duet vector, respectively. “Sofic + AS1” indicates that *EcSofic* was cloned into the first MCS, while *EcAS1* was cloned into the second MCS. Conversely, “AS1 + Sofic” signifies that *EcAS1* was cloned into the first MCS and *EcSofic* into the second MCS.

**Figure 2 ijms-26-06074-f002:**
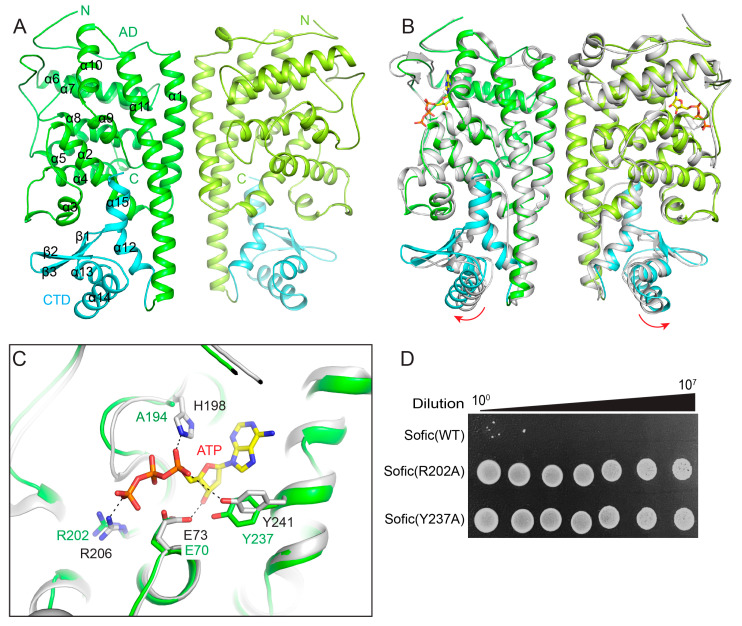
Structure of the *Ec*Sofic dimer. (**A**) Ribbon diagram of the crystal structure of the *Ec*Sofic dimer. The N- and C-termini of *Ec*Sofic are indicated, and the secondary structures are labeled. In this drawing, the AMPylase domain (AD) is colored in green and lime, while the C-terminal domain (CTD) is colored in cyan. (**B**) Superimposed structures of apo *Ec*Sofic and ATP-bound Sofic from *Shewanella oneidensis* MR-1(*So*Sofic). *Ec*Sofic is colored as in panel A, while *So*Sofic is colored in gray. (**C**) Close-up view of the ATP-binding pockets of the overlaid structures of *Ec*Sofic and *So*Sofic. The key residues required for ATP binding are labeled and shown in sticks, and the salt bridges and hydrogen bonds for ATP stabilization are indicated by the black dashed line. (**D**) Colony-forming assays evaluating the toxic effects of *Ec*Sofic and its mutants.

**Figure 3 ijms-26-06074-f003:**
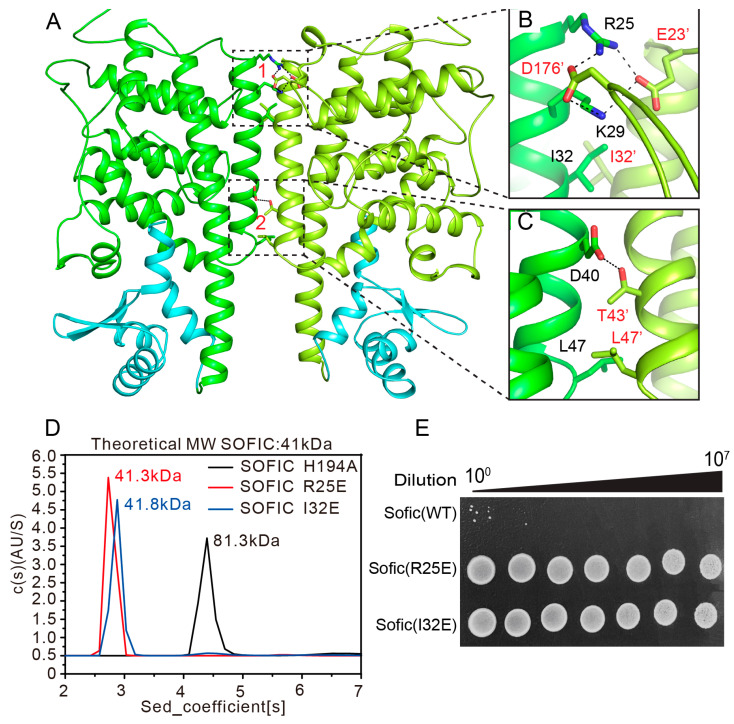
Dimerization is critical for Sofic’s toxic activity. (**A**) Ribbon diagram of the overall structure of the *Ec*Sofic dimer. The two major interfaces mediating the dimerization are boxed in by the dash line and labeled. (**B**,**C**) Close-up views of the two interfaces required for dimerization are shown: interface 1 in panel B and interface 2 in panel C. Key residues involved in mediating Sofic dimer formation are shown as sticks, and the hydrophilic networks are indicated by black dashed lines. (**D**) Analytical ultracentrifugation: the molecular weights of the *Ec*Sofic H194A, R25E, and I32E mutants. (**E**) Colony-forming assays showing the toxic effects of *Ec*Sofic R25E and I32E mutants.

**Figure 4 ijms-26-06074-f004:**
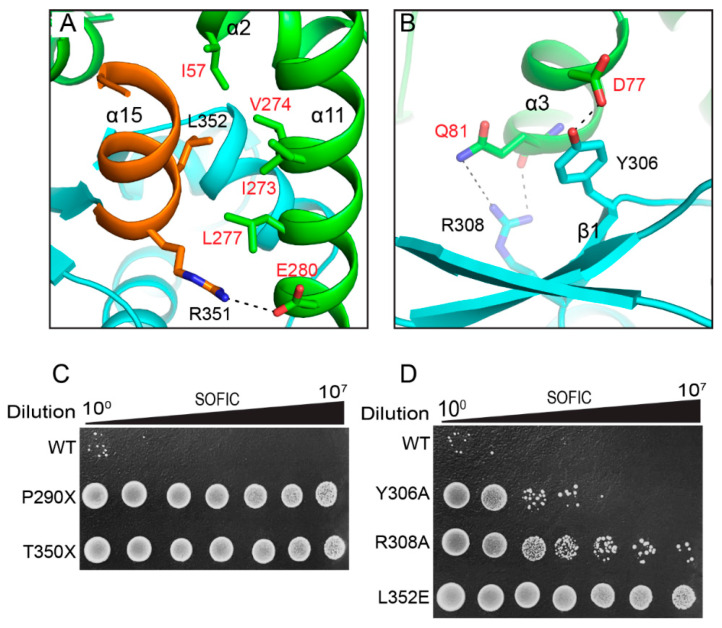
The toxic activity of Sofic is regulated by its CTD. (**A**) Close-up view of the interactions between CTD α15 and α11 of the AMPylase domain. (**B**) Close-up view of the interactions between β1 of the CTD and α3 of the AMPylase domain. (**C**,**D**) Colony-forming assays analyzing the toxic effects of truncated *Ec*Sofic or its mutants.

**Table 1 ijms-26-06074-t001:** Diffraction data collection and refinement statistics.

Structure	*Ec*Sofic
**PDB ID**	9VJC
**Data collection**	
Space group	P1211
Cell dimensions	
a, b, c (Å)	68.595, 58.234, 102.863
α, β, γ (º)	90, 93.57, 90
Resolution (Å)	55.39–2.283 (2.364–2.283) *
*R*_pim_ (%)	6.0 (28.0)
*I/σI*	11.5 (3.0)
CC (1/2)	0.995(0.922)
Completeness (%)	94.36 (86.80)
**Refinement**	
No. of reflections	35,092 (3222)
*R_work_*/*R_free_ *(%)	0.2117 (0.2767)/0.2666 (0.3527)
Ramachandran	
Favored (%)	99.72
Outlier (%)	0.28
R. m.s. deviations	
Bond lengths (Å)	0.008
Bond angles (°)	0.97
Average *B*-factor	31.09
Protein	30.90
Solvent	34.26
Number of non-hydrogen atoms	6125
Protein	5774
Ligand	0
Solvent	351

* Highest resolution shell shown in parentheses.

## Data Availability

The coordinate and structure factors of the crystal structure of *Ec*Sofic H194A have been deposited into the Protein Data Bank with the accession code 9VJC. All other data are available from the corresponding author upon request.

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
