# Peer review of "Structural Insights into the Regulatory Mechanisms of the Toxic Activity of Sofic in Anti-Phage Defense Systems"

_ijms, 2025, doi:10.3390/ijms26136074_

Round 1

Reviewer 1 Report

Comments and Suggestions for Authors

The research topic on toxin-antitoxin defense mechanisms in bacteria against bacteriophage infection is relevant, as it could provide valuable information for improving the use of bacteriophages as antimicrobials, a good alternative to antibiotic resistance. However, the document requires significant improvements in its wording to ensure a clear understanding of the results obtained and their explanations.

The attached document provides some clarifications on the sections that require improvement. Italics should also be used throughout the document, and discussions should be reinforced with scientific references.

Reviewer 2 Report

Comments and Suggestions for Authors

The study primarily focuses on the bacterial strategies against phage infection.

Does it contribute towards bacterial abilities to develop resistance against phage infection?

What is the range of distribution in bacteria?

Are there phages which can overcome bacterial toxicity?

How can phages acquire immunity against Sofic protein?

A few queries related to the molecular mechanisms regulating Sofic protein's toxic activity which protects bacteria from viral infection

How many genes encode this Sofic protein's toxic activity?

Are there any major differences in the number and size of genes in prokaryotes and eukaryotes?

What are the phage-specific molecular cues, which activate the phage signals?

What is the mode of toxicity of Sofic? Is this expression related to / act in combination with any other factor?

How can phage evade the Sofic’s toxicity?

What is the location of  sofiC genes with respect to other phage defense systems

What are the evolutionary significance of these system?

Reviewer 3 Report

Comments and Suggestions for Authors

Comments

The work of Wu Zhuoxi, et al. described the regulatory mechanisms of Sofic, a toxic protein that has shown importance during phage invasion. The introduction is clear, presenting the state of the art and the rational behind the work. The results are presented in a very comprehensive way and described sufficiently. The discussion must be improved. The Materials and Methods has the necessary information. Overall, the manuscript is interesting, it is well-written, and all the conclusions are supported by the results. Thus, after the changes recommended, the reviewer suggests its publication on IJMS since it fits into the scope of the journal.

Major comments:

  1. The number of the figures in the text must appear sequentially.
  2. The discussion must have more details and a clear correlation with the results presented.
  3. Do not present the results as dilution, but as concentration or molar concentration.

Minor comments:

Line 17 – In vivo?

Line 24 – antipage or anti-phage?

Line 41 – Fic or FIC?

Line 63 – Why did you select E coli? Write a brief sentence on the reason behind the selection of this model.

Line 75 – Which are the implications of using a different Sofic? Briefly state the limitations of using Sofic from other source.

Line 79 – Why do the authors call in vivo assay if they are using cells?

Line 84 – Where is the reference to figure 1A? Authors must follow the order of the fugures in the text.

Line 99 – Figure 1. Do you have a negative control that did not induce any alterations even at higher concentrations? Can you quantify the concentration of Sofic, instead of representing the dilution?

Line 152 – Figure 3D before Figure 3A, B, and C…

Line 247 – Did you quantify the concentration? The results must be expressed as X concentration and Y effect instead of the dilution.

Line 281 – This is an in vitro assay, not in vivo

Round 2

Reviewer 2 Report

Comments and Suggestions for Authors

Accept

Reviewer 3 Report

Comments and Suggestions for Authors

The authors replied successufully to all my comments